# Unprotected Left Main Bifurcation Stenting in Acute Coronary Syndromes: Two-Stent Technique versus One-Stent Technique

**DOI:** 10.3390/jpm13040670

**Published:** 2023-04-16

**Authors:** Lucian Predescu, Marin Postu, Lucian Zarma, Adrian Bucsa, Pavel Platon, Marian Croitoru, Adrian Mereuta, Leonard Licheardopol, Alexandra Predescu, Dan Dorobantu, Dan Deleanu

**Affiliations:** 1Cardiology Department, “Prof. CC Iliescu” Institute for Cardiovascular Diseases, 022328 Bucharest, Romania; marin_postu@yahoo.com (M.P.); lucianzarma001@yahoo.com (L.Z.); abucsa@yahoo.com (A.B.); pavel_platon@yahoo.com (P.P.); drcroitoru@gmail.com (M.C.); admereuta@yahoo.com (A.M.); dn.dorobantu@gmail.com (D.D.); dan.deleanu@yahoo.com (D.D.); 2Faculty of Medicine, “Carol Davila” University of Medicine and Pharmacy, 050474 Bucharest, Romania; 3Cardiology Department, Tulcea County Hospital, 820195 Tulcea, Romania; licheardopol@gmail.com; 4Doppler Clinic, 030136 Bucharest, Romania; alexandraiernici@yahoo.com; 5Faculty of Health Sciences, University of Bristol, Bristol BS8 1QU, UK

**Keywords:** left main coronary lesion, acute coronary syndrome, two-stent technique

## Abstract

Aims: There is little evidence guiding the choice between a one-stent and a two-stent approach in unprotected distal left main coronary artery disease (UDLMCAD) presenting as acute coronary syndrome (ACS). We aim to compare these two techniques in an unselected ACS group. Methods and results: We conducted a single center retrospective observational study, that included all patients with UDLMCAD and ACS undergoing PCI between 2014 and 2018. Group A underwent PCI with a one-stent technique (*n* = 41, 58.6%), Group B with a two-stent technique (*n* = 29, 41.4%). A total of 70 patients were included, with a median age of 63 years, including *n* = 12 (17.1%) with cardiogenic shock. There were no differences between Group A and B in terms of patient characteristics, including SYNTAX score (median 23). The 30-day mortality was 15.7% overall, and was lower in Group B (3.5% vs. 24.4%, *p* = 0.02). Mortality rate at 4 years was significantly lower in Group B (21.4% vs. 44%), also when adjusted in a multivariable regression model (HR 0.26, *p* = 0.01). Conclusions: In our study, patients with UDLMCAD and ACS undergoing PCI using a two-stent technique had lower early and midterm mortality compared to one-stent approach, even after adjusting for patient-related or angiographic factors.

## 1. Introduction

Developments from past years in stent technology and the use of intravascular imaging to assess the results after percutaneous coronary interventions (PCIs) led to a rapid increase in the number of patients with unprotected left main coronary artery disease (UDLMCAD) treated by PCI. Many studies have excluded patients with acute coronary syndrome (ACS) [1]. Therefore, there is a significant gap in knowledge regarding the treatment of patients with UDLMCAD in acute settings. There is little evidence guiding the choice between one-stent and two-stent approaches in UDLMCAD presenting as ACS [1,2,3,4,5,6,7,8,9,10,11,12,13,14,15]. We aimed to compare these two techniques in an unselected ACS group.

The aim of our study was to define the current practice for patients with ACS and culprit left main lesions treated by primary PCI, in a Romanian high-volume PCI center and compare its outcomes with those reported by other studies. One-stent technique was compared with two-stent technique in patients with UDLMCAD presented as ACS and treated by PCI.

## 2. Methods

### 2.1. Study Population

Between January 2014 and December 2018, a total of 146 cases of PCI in unprotected left main coronary disease were performed. Of these, 76 cases were left main primary PCI in patients presented with ACS. In 70 cases the bifurcation of left main was involved, this latter figure representing the study population. All the patients were refused by heart surgeons due to emergency presentation and prior antiplatelet therapy. In all cases the choice of technique, type of stent and perioperative care was decided by the attending physician(s), as per current guidelines and local protocols. For all patients the electronic hospital records were reviewed, including angiography and angioplasty imaging. All clinical or angiographic characteristics were defined and classified as per current practice and guidelines [11,12,14]. Follow up was through in-hospital records (*n* = 38, 54.2%) or a combination of telephone interview by attending physician and the National Insurance Agency Platform (*n* = 32, 45.7%). The cause of death was not documented or clear from available data in 5 out of 24 cases, and as such the outcome of all-cause mortality, instead of cardiovascular mortality, was used, to avoid confusion.

### 2.2. Study Outcomes

Major adverse cardiac events (MACEs) were defined as the occurrence of death (all-cause mortality), myocardial infarction (MI) or target lesion revascularization (TLRs). ACS was defined as either unstable angina, non-ST-segment elevation MI or ST segment-elevation MI. TLR was defined as repeated PCI for restenosis of the entire segment involving the implanted stent and the 5-mm distal and proximal borders adjacent to the stent. Stent thrombosis was defined based on Academic Research Consortium definitions according to timing of presentation as early (0–30 days), late (31–360 days), or very late (>360 days). Angiographic success was defined as residual stenosis of <30% by visual estimation in the presence of Thrombosis in Myocardial Infarction (TIMI) flow grade 3. Complete revascularization was defined as any attempt to revascularize all diseased segments (≥2.5 mm in diameter). The diagnosis of periprocedural MI was made if after PCI there was an increase in CK-MB or troponin levels that was 5 times the upper normal level [11,12,14].

### 2.3. Statistical Analysis

Frequencies are given as numbers and percentages, continuous values as median (inter-quartile range or minimum-maximum values). Population characteristics were compared using the Mann–Whitney U test, Kruskall–Wallis test and Fisher’s exact test. Patients were divided into two main groups based on the PCI technique used: Group A—one-stent technique and Group B—two-stent technique [11,12,14].

Early outcomes (mortality, stent thrombosis, need for mechanical circulatory support, access site complications) are based on status at 30 days and presented as percentages. Late outcomes are estimated using the Kaplan–Meier method. Late outcomes of interest are all-cause mortality, TLR and MACE [11,12,14].

Predictors of early outcomes were identified using univariable linear regression adjusted by cardiogenic shock at the time of procedure. Multivariable analysis was not possible due to only 11 early events. Predictors of late outcomes death were identified using a combination of backward and forward stepwise multivariable Cox regression, including all variables with a univariable regression *p* value of less than 0.1 and less than <10% missing values. The statistically significant variables left in the final model were considered as independent predictors. The Group A vs. Group B variable was always kept in the model, as the variable of interest. With only 6 TLR events, only univariable analysis was performed [11,12,14].

Statistical analyses were performed with STATA/SE 12.0 (StataCorp LP, College Station, TX, USA).

## 3. Results

### 3.1. Patient Population

A total of 70 patients undergoing distal left main PCI in ACS were included, with ages ranging from 33 to 86 years (median of 64.5 years). Group A consisted of *n* = 41 (58.6%), while Group B included the remaining *n* = 29 (41.4%). Detailed demographic, baseline clinical characteristics and risk scores by group are presented in Table 1, with no statistically significant differences between the two groups.

### 3.2. Angiographic Characteristics

Table 2 shows the main angiographic findings by used technique. As expected, true bifurcation lesions (Medina 1/1/1 and 0/1/1) were more frequent in Group B, as were those involving the left circumflex artery (LCX) (1/0/1 and 0/0/1). Medina 0/1/0 or those involving the left anterior descending artery (LAD) (1/0/0, 1/1/0) were more frequent in Group A. Left main bifurcation angles >90 degrees were more frequent in Group A.

### 3.3. Procedural Data

Of the 43 patients with a myocardial infarction, *n* = 31 (81.4%) underwent PCI at the same time as the diagnostic angiography, while *n* = 8 (28.6%) underwent PCI following 2 to 6 days (*n* = 6 in NSTEMI, *n* = 2 in STEMI). Of the 27 patients with unstable angina, *n* = 10 (37%) underwent PCI at the same time as the diagnostic angiography, *n* = 9 (33.3%) underwent PCI during the same hospitalization, while the remainder of *n* = 8 patients (29.7%) had an initial medical control of the angina after the diagnostic angiography and returned later with an episode of unstable angina requiring PCI.

Six patients (8.6%) were on a mechanical support system with IABP prior to the PCI procedure. Femoral access was the preferred option, in 92.9% of cases. Either a 6F (60%) or a 7F (38.6%) guiding catheter was used in most procedures.

A variety of stents were used, the most common being Stentys (STENTYS SA, Paris, France) with *n* = 32 (45.7%), Biomime (Meril Life Sciences Pvt. Ltd., Gujarat, India) with *n* = 14 (20%) and Xience (Abbott Vascular, Santa Clara, California) with *n* = 15 (21.4%). More than one type of stent was implanted in *n* = 15 patients (21.4%), more frequently in Group B. There was a trend for more frequent use of Stentys stents in Group A and Xience stents in Group B.

In Group B the two-stent techniques used were: provisional T stenting (3.5%), T and small protrusion (TAP) (35%), minicrush (31%), double kissing crush (3.5%), culotte (27%).

More details on the procedural steps, use of proximal optimization technique (POT) and kissing balloon post-dilatation (KBPD) are shown in Table 3. In most Group A cases the first stented vessel was the LAD, while in Group B it was the CX. POT was used in similar proportions in both groups (53.7% in Group A vs. 69% in Group B, *p* = 0.2), but was performed after KBPD more frequently in Group B (9.8% in Group A vs. 41.4% in Group B, *p* = 0.002), owing in part to the higher use of KBPD in this group (22.5% in Group A vs. 75% in Group B, *p* < 0.001).

Technical outcomes and post-procedural complications by strategy group are summarized in Table 4. The single stent technique (Group A) resulted in more side branch residual stenosis, fewer cases of successful PCI, complete revascularization and TIMI 3 flow at the end of the procedure. The 8 cases with TIMI < 3 were anterior STEMI with severe systolic left ventricular dysfunction in 7/8 cases and cardiogenic shock in 5/8 cases. There were no differences between the two groups in terms of periprocedural complications.

### 3.4. Early Outcomes

Early mortality (30 days mortality) was 15.7% (*n* = 11), higher in Group A compared to Group B (24.4% vs. 4.4%, *p* = 0.02), with a 4.3% peri-procedural mortality (*n* = 3).

Most deaths occurred in patients presenting with cardiogenic shock (*n* = 8). Early mortality in those not presenting as cardiogenic shock was 5.2% (*n* = 3) with 1.7% (*n* = 1) periprocedural deaths.

Three patients underwent an emergency angiographic reevaluation during the same hospitalization, 2 in Group A and one in Group B. In two cases, an acute in stent thrombosis was found (one in Group A, one in Group B), while in the third case an acute LCX thrombosis was found, with permeable left main stent (in Group A). All three patients had presented initially with MI and died during the same hospitalization.

Post-procedural IABP was used in 8 cases (n = 8.6%, no significant differences between Group A and B). We found that in patients with cardiogenic shock at presentation pre-procedural IABP was not associated with decreased early mortality (60% vs. 71%, *p* = 0.6), but use of IABP for post-procedural support was associated with a decrease in early mortality (20% vs. 100%, *p* = 0.01).

Predictors of early mortality are summarized in Table 5. All values are from bivariable regression adjusted by presence of cardiogenic shock since most deaths occurred in these patients. Cardiogenic shock itself is a strong predictor of early mortality, with an HR of 13.6 (*p* = 0.001). The one stent technique (Group A) remained a strong predictor of early mortality even when adjusted for cardiogenic shock (HR 0.03, *p* = 0.02), but not when adjusted by TIMI < 3. Nevertheless, there is a trend for higher mortality in Group A with TIMI flow 3 compared to Group B with TIMI flow 3 (12.2% vs. 3.4%, *p* = 0.2).

### 3.5. Late Outcomes

Mortality, TLR and MACE at 4 years were 34.6%, 13.6% and 41.5% overall. Unadjusted comparisons of mortality, TLR and MACE at 4 years between Group A and Group B are shown in Figure 1. Mortality rate at 4-year follow up remained lower in Group B compared to Group A after multivariable analysis (HR 0.26, *p* = 0.01).

Predictors of late mortality and coronary reintervention are summarized in Table 6. Current risk and complexity scores were not independent predictors of mortality at 4 years in this group of ACS patients, but cardiogenic shock, hemoglobin under 12 g/dL, severe systolic left ventricular dysfunction and diffusely infiltrated left main were.

To evaluate whether the post-procedural TIMI < 3 acted as a confounder due to bias of treatment choice (less frequent use of two stent technique when TIMI flow is not satisfactory during the procedure), this variable was forced into the model, with minimal change to results. In addition, we performed a sensitivity analysis by testing the model in the subgroup of patients where the TIMI flow at the end of the procedure was 3, and the two-stent technique remained associated with lower mortality at 4 years (HR 0.32, *p* = 0.05). To further evaluate the interaction between technique used and TIMI flow achieved, a variable was defined with the following categories: Group B, TIMI flow 3, Group A, TIMI flow 3 and Group A TIMI flow < 3. This variable was introduced in the previous model, replacing Group B vs. Group A. We found that Group A with TIMI flow < 3 was associated with a 6.5 higher risk of death at 3 years compared to Group A and TIMI flow 3 (*p* = 0.006) while Group A with TIMI flow of 3 was associated with a 3.1 times higher risk of death at three years compared to Group B and TIMI flow 3 (*p* = 0.04).

Diabetes and a history of atrial fibrillation were independent predictors of TLR at 4 years. Since most events in the composite MACE were deaths (20/26), predictor analysis was not repeated for this outcome.

## 4. Discussion

In this observational study comparing a one-stent to a two-stent strategy in patients with ULMCAD presenting as ACS, we found that a two-stent technique was associated with lower early and 4-year all-cause mortality, after adjusting for patient clinical and angiographic characteristics (HR 0.26, *p* = 0.01).

There are conflicting data on the best PCI strategy for UDLMCAD, especially when considering newer techniques such as the DK-crush. Some studies have shown that a one-stent strategy was associated with a reduction of long-term all-cause mortality compared to a two-stent strategy in these patients [4,5]. Other studies have shown similar outcomes for both techniques [6,7,8,9]. Although without any impact on mortality rate, the double kissing crush technique reduced the rate of target lesion failure at 1 year compared to provisional stenting [10]. Nevertheless, the generally accepted consensus at the moment is that a single stent strategy is the preferred strategy. When it comes to the subgroup presenting as ACS, however, data are lacking, as these patients are either excluded from these studies, or not analyzed separately; to our knowledge, there are no reports specifically addressing this subgroup of patients undergoing PCI.

We identified no significant differences in clinical and demographic characteristics between the two groups. In 41.4% of patients, there was a need to implant another stent to treat the distal left main bifurcation. As expected, true bifurcation lesions were treated more often with a two-stent strategy. We identified a lower use of POT (60%) and KBPD (44.1%) in our study. KBPD was more frequently performed when a two-stent strategy was used (22.5% vs. 75%). This reflects the tendency to keep the PCI procedure as simple as possible in an acute setting. No harm was detected when two-stent strategy was implemented when needed.

We observed that all patients with a TIMI flow of 1 or 2 were in the single stent group, and this can be explained by a reluctance in switching to a more complex procedure when poor distal flow is observed during the PCI in an acute, critical setting. To determine whether the results were biased by this finding, extensive sensitivity analysis was performed, and we found that the lower mortality for a two-stent strategy was maintained even when adjusting for TIMI flow.

There were two cases of acute instent thrombosis and one case of acute side branch vessel thrombosis with a fatal outcome. In cases of acute instent thrombosis, no precipitating mechanical factors have been identified for it such as underexpansion or nonapposition of the stent. The most likely cause of these acute insistent thrombosis cases is inadequate antiplatelet therapy. In the case of acute thrombosis at the level of the circumflex artery after left main PCI, we noticed that KBPD was not performed, which could be the cause of the thrombosis.

In an acute setting some factors can alter the outcomes, not found in elective cases. In ACS, local thrombosis is favored by ruptured plaque, high platelet reactivity and post-procedural small local dissection. This scenario can happen especially with a one-stent technique after KBPD when the ostium of the side branch can suffer a small injury. Because in ACS there is a high local thrombotic environment, these small side branch injuries can increase the risk for thrombosis. We speculate that, in patients with ACS and UDLMCAD, ensuring a good angiographic result without overlooking small dissection and residual side branch stenosis could have an important role, even if this means switching to a two-stent strategy.

Additionally, there is the role played by ischemic areas and residual lesions. Most patients who died at 4 years had decreased LVEF, and it can be argued that minimizing residual ischemia to non-infarcted areas, by side branch stenting, can have a role in preserving or recovering left ventricular function, both early and late.

## 5. Limitations

This study was a non-randomized study in which operator bias may have influenced the conclusion. Although it might have offered more data on restenosis, routine angiographic reevaluation is no longer recommended and was not performed. The small number of early events prevented multivariable analyses, so the results are subject to confounding. The use of various types of stents introduced a degree of heterogeneity.

## 6. Conclusions

Patients with UDLMCAD and ACS undergoing PCI using a two-stent technique had lower early and midterm mortality compared to one-stent approach, even after adjusting for patient-related or angiographic factors. This finding could change the approach during PCI of left main stenosis in an acute setting. A two-stent approach might offer better early perfusion of both the main vessel and of the side branch, possibly contributing to better recovery of myocardial function and fewer deaths due to progression of heart failure.

## Figures and Tables

**Figure 1 jpm-13-00670-f001:**
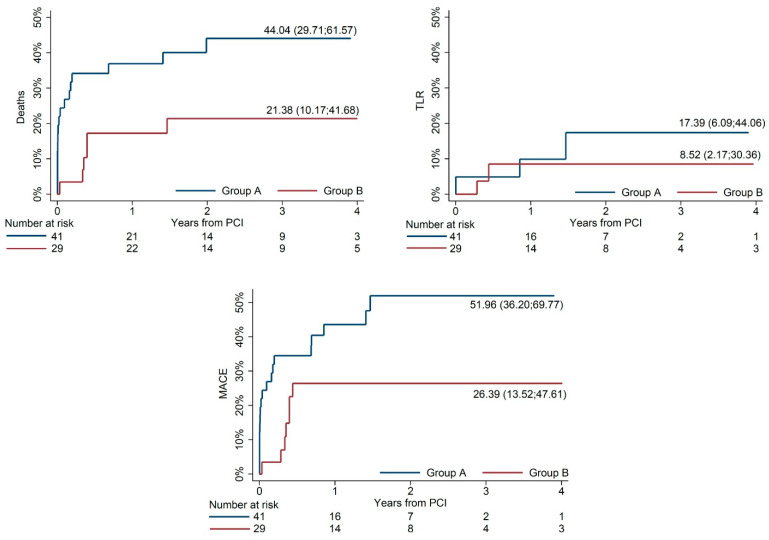
Unadjusted comparisons of mortality, TLR and MACE at 4 years according to the stent strategy in patients with unprotected left main coronary artery disease and acute coronary syndrome (Group A—one-stent strategy; Group B—two-stent strategy). Curves represent Kaplan–Meier failure function. TLR—target lesion revascularization; MACE—major adverse cardiac event.

**Table 1 jpm-13-00670-t001:** Demographic and baseline clinical characteristics according to the stent strategy in patients with unprotected left main coronary artery disease and acute coronary syndrome (Group A—one stent strategy; Group B—two stent strategy).

	Group A	Group B	Total	*p* Value
	*n* = 41	*n* = 29	*n* = 70	
Age, y (median, IQR)	63 (52;71)	69 (57;72)	64.5 (57;72)	0.4
Male	30 (73.2)	20 (69)	50 (71.4)	0.8
Cardiovascular risk factors				
Hypertension	31 (75.6)	26 (89.7)	57 (81.4)	0.2
Dyslipidemia	32 (80)	23 (79.3)	55 (79.7)	0.6
Diabetes mellitus	10 (25)	11 (37.9)	21 (30.4)	0.3
Obesity	8 (19.5)	8 (27.6)	16 (22.9)	0.6
Active smoker	15 (36.6)	4 (13.8)	19 (27.1)	0.1
Former smoker	5 (12.2)	5 (17.2)	5 (17.2)	0.1
Patient history				
ACS	10 (25)	9 (31)	19 (27.5)	0.6
Unstable angina	1 (10)	3 (33.3)	4 (21.1)	0.6
NSTEMI	4 (40)	3 (33.3)	7 (36.8)	
STEMI	5 (50)	3 (33.3)	8 (42.1)	
PCI	6 (12.8)	8 (27.6)	14 (18.4)	0.1
Atrial fibrillation	6 (15)	4 (13.8)	10 (14.5)	0.6
Stroke/TIA	2 (4.9)	5 (17.2)	7 (10)	0.1
Bleeding	0 (0)	1 (3.5)	1 (1.5)	0.4
COPD	0 (0)	0 (0)	0 (0)	
Peripheral artery disease	6 (15)	1 (3.5)	7 (10.1)	0.2
Neoplasia	0 (0)	3 (10.3)	3 (4.5)	0.07
Type of ACS at presentation				
Unstable angina	13 (31.7)	14 (48.3)	27 (38.6)	0.2
NSTEMI	6 (14.6)	7 (24.1)	13 (18.6)	
STEMI	14 (34.1)	4 (13.8)	18 (25.7)	
ACS with cardiogenic shock	8 (19.5)	4 (13.8)	12 (17.1)	
Clinical characteristics at presentation				
Atrial fibrillation	5 (12.8)	3 (10.3)	8 (11.8)	0.5
Left bundle branch block	3 (7.7)	5 (17.2)	8 (11.8)	0.3
Right bundle branch block	5 (12.8)	1 (3.5)	6 (8.8)	0.2
Ventricular tachycardia	2 (5.1)	0 (0)	2 (2.9)	0.5
LVEF, % (median, IQR)	55 (50–60)	55 (45–60)	55 (45–60)	0.3
LV systolic dysfunction				
None (≥50%)	15 (36.6)	13 (44.8)	28 (40)	0.4
Mild (40–49%)	8 (19.5)	6 (20.7)	14 (20)	
Moderate (30–39%)	7 (17.1)	7 (24.1)	14 (20)	
Severe (<30%)	11 (26.8)	3 (10.3)	14 (20)	
Regional wall motion abnormality	34 (82.9)	20 (69)	54 (77.1)	0.2
Blood samples pre-PCI				
Hb, g/dL (median, IQR)	14.2 (11.9;15.2)	13 (12.3;14.4)	13.6 (11.9;15)	0.2
CK-MB, U/L (median, IQR)	36 (19;77)	33 (19;125)	33 (19;92)	0.8
Troponin *, ng/mL (median, IQR)	0.2 (0.03;1.4)	0.9 (0.09;7.2)	0.5 (0.03;3)	0.3
Creatinine clearance, mL/min/1.73 m^2^ (median, IQR)	78 (47;95)	81 (59;99)	80 (48;97)	0.5
Angiographic and clinical risk scores				
SYNTAX	32 (12;32)	25 (17;30)	23.2 (15;32)	0.6
SYNTAX-2 PCI	39.1 (25.4;51.7)	32.5 (26.3;43.3)	34.5 (25.9;50.1)	0.4
Estimated PCI 4-year mortality	14 (4.7;35.4)	9 (5.8;19.5)	10.6 (5;31.8)	0.6
SYNTAX-2 CABG	29.8 (19.3;41.1)	31.4 (22.4;36.4)	30.6 (21.1;37.5)	0.8
Estimated CABG 4-year mortality	6.7 (2.8;16.8)	7.6 (3.6;11.4)	7.2 (3.3;12.6)	0.8
EUROSCORE II	2.7 (1.1;5.9)	1.3 (1.1;4.2)	2.5 (1.1;5.2)	0.3

ACS—acute coronary syndrome; COPD—chronic obstructive pulmonary disease; EUROSCORE—European System for Cardiac Operative Risk Evaluation; Hb—hemoglobin; NSTEMI—non-ST-segment elevation myocardial infarction; LVEF—left ventricular ejection fraction; PCI—percutaneous coronary intervention; STEMI—ST-segment elevation myocardial infarction; TIA—transient ischemic attack; SYNTAX—Synergy between PCI with TAXUS drug-eluting stent and cardiac surgery. * In 32 patients with troponin measurements.

**Table 2 jpm-13-00670-t002:** Angiographic findings of patients with unprotected left main coronary artery disease and acute coronary syndrome (Group A—one stent strategy; Group B—two stent strategy).

	Group A	Group B	Total	*p* Value
	*n* = 41	*n* = 29	*n* = 70	
Arterial access site				
Radial	4 (9.8)	4 (13.8)	8 (11.4)	0.7
Femoral	37 (90.2)	25 (86.2)	62 (88.6)	0.7
LM lesion type				
Distal	35 (85.4)	22 (75.9)	57 (81.4)	0.3
Ostial and distal	1 (2.4)	0 (0)	1 (1.4)	
Whole length	5 (12.2)	7 (24.1)	12 (17.2)	
Bifurcation	30 (73.2)	26 (89.7)	56 (80)	0.1
Trifurcation	11 (26.8)	3 (10.3)	14 (20)	0.1
Other coronary lesions				
None	19 (46.3)	10 (34.5)	29 (41.4)	0.7
One vessel	10 (24.4)	10 (34.5)	20 (28.6)	
Two vessels	8 (19.5)	6 (20.7)	14 (20)	
Three vessels	4 (9.8)	3 (10.3)	7 (10)	
Chronic total obstruction	8 (19.5)	3 (10.3)	11 (15.7)	0.3
LAD ostium involved	32 (78.1)	23 (79.3)	55 (78.6)	0.6
LCX ostium involved	9 (21.9)	22 (75.9)	31 (44.3)	<0.001
LAD non-ostial lesion	16 (39)	12 (41.4)	28 (40)	0.5
LCX non-ostial lesion	5 (12.2)	11 (37.9)	16 (22.9)	0.02
RCA lesion	11 (26.8)	6 (20.7)	17 (24.3)	0.6
LM lesion characteristics				
Diffuse lesion	18 (43.9)	17 (58.6)	35 (50)	0.3
Eccentric lesion	30 (73.2)	21 (72.4)	51 (72.9)	0.6
Calcified lesion	13 (31.7)	13 (44.8)	26 (37.1)	0.3
Ulcerated lesion	22 (52.7)	10 (34.5)	32 (45.7)	0.2
Carina involvement	6 (14.6)	2 (6.9)	8 (11.4)	0.5
Medina classification				
1/1/1	7 (17.1)	12 (41.4)	19 (27.1)	<0.001
1/0/0	7 (17.1)	1 (3.5)	8 (11.4)	
1/1/0	10 (24.4)	4 (13.8)	14 (20)	
1/0/1	0 (0)	3 (10.3)	3 (4.3)	
0/1/0	15 (36.6)	2 (6.9)	17 (24.3)	
0/1/1	1 (2.4)	5 (17.2)	6 (8.6)	
0/0/1	1 (2.4)	2 (6.9)	3 (4.3)	
True bifurcation (1/1/1, 0/1/1)	8 (19.5)	17 (58.6)	25 (35.7)	0.001
LM take-off angle >70 degrees	14 (38.9)	9 (37.5)	23 (38.3)	0.8
Bifurcation angle				
>90 degrees	7 (21.2)	1 (4)	8 (13.8)	0.05
70–90	15 (45.5)	8 (32)	23 (39.7)	
45–69	3 (9.1)	8 (32)	11 (19)	
<45	8 (24.2)	8 (32)	16 (27.5)	
LM stenosis, % (median, IQR)	55.5 (10;95)	55 (23;90)	55 (20;90)	0.8
LAD stenosis, % (median, IQR)	90 (72;97)	85 (78;92)	87 (75;93)	0.2
LCX stenosis, % (median, IQR)	73 (20;92)	75 (50;90)	75 (43;90)	0.6

LCX—left circumflex artery; LM—left main; LAD—left anterior descending artery; RCA—right coronary artery; LM take-off angle—the angle between the left main and the sinus of Valsalva.

**Table 3 jpm-13-00670-t003:** Procedural characteristics according to the stent strategy in patients with unprotected left main coronary artery disease and acute coronary syndrome (Group A—one stent strategy; Group B—two stent strategy).

	Group A	Group B	Total	*p* Value
	*n* = 41	*n* = 29	*n* = 70	
Arterial access site				
Femoral	38 (92.7)	27 (93.1)	65 (92.9)	0.6
Radial	3 (7.3)	2 (6.9)	5 (7.1)	0.6
Pre-PCI IABP	3 (7.3)	3 (10.3)	6 (8.6)	0.7
Guide catheter				
6F	30 (73.2)	12 (41.4)	42 (60)	0.01
7F	10 (24.4)	17 (58.6)	27 (38.6)	
8F	1 (2.4)	0 (0)	1 (1.4)	
Rotational atherectomy	1 (2.4)	0 (0)	1 (1.4)	0.6
Stent type (main 3)				
Stentys	22 (53.7)	10 (34.5)	32 (45.7)	0.2
Xience	5 (12.2)	10 (34.5)	15 (21.4)	0.04
Biomime	8 (19.5)	6 (20.7)	14 (20)	0.6
First stented vessel				
LAD	39 (97.5)	10 (34.5)	49 (71)	<0.001
CX	1 (2.5)	19 (65.5)	20 (29)	
MV predilatation	30 (75)	21 (72.4)	51 (73.9)	0.5
SB predilatation	7 (17.5)	17 (58.6)	24 (34.8)	0.001
Predilatation at nominal diameter *	10 (33.3)	9 (42.9)	19 (37.3)	0.6
Dissection after predilatation *	6 (20)	10 (47.6)	16 (31.4)	0.06
POT	22 (53.7)	20 (69)	42 (60)	0.2
First POT—after stent implantation	19 (46.3)	16 (55.2)	35 (50)	0.6
Second POT—after KBPD	4 (9.8)	12 (41.4)	16 (22.9)	0.002
KBPD	9 (22.5)	21 (75)	30 (44.1)	<0.001
TKBPD	0 (0)	2 (6.9)	2 (2.9)	0.2
MV stent diameter				
3 mm	9 (22)	7 (24.1)	16 (22.9)	0.6
3.5 mm	28 (69.3)	21 (72.4)	49 (70)	
4 mm	4 (9.8)	1 (3.5)	5 (7.1)	
MV stent length, mm (median, IQR)	22 (22;27)	27 (19;29)	23 (22;28)	0.2
SB stent diameter, mm (median, IQR)		3.5 (3;3.5)		
SB stent length, mm (median, IQR)		19.5 (13;23)		
Under expansion of >30%	7 (17.1)	3 (10.3)	10 (14.3)	0.5
IFR used before PCI	0 (0)	1 (3.5)	1 (1.4)	0.4
IFR used after PCI	0 (0)	1 (3.5)	1 (1.4)	0.4
IVUS used after PCI	3 (7.3)	7 (24.1)	10 (14.3)	0.08

* Only when MV predilation was used. BES—biolimus eluting stent; CX—left circumflex artery; EES—everolimus eluting stent; IABP—intra-aortic balloon pump; iFR—instantaneous wave-free ratio; IVUS—intravascular ultrasound; KBPD—kissing balloon post-dilatation; LAD—left anterior descending artery; MV—main vessel; POT—proximal optimization technique; PCI—percutaneous coronary intervention; SES—sirolimus eluting stent; SB—side branch; TKBPD—triple kissing balloon post-dilatation; ZES—zotarolimus eluting stent.

**Table 4 jpm-13-00670-t004:** Procedural outcomes and complications in patients with unprotected left main coronary artery disease and acute coronary syndrome (Group A—one stent strategy; Group B—two stent strategy).

	Group A	Group B	Total	*p* Value
	*n* = 41	*n* = 29	*n* = 70	
Unaffected LM ostium covered	4 (9.8)	5 (17.2)	9 (12.9)	0.5
LM lesion covered entirely	39 (95.1)	29 (100)	68 (97.1)	0.5
SB residual stenosis				
None	19 (47.5)	22 (75.9)	41 (59.4)	0.01
<50%	12 (30)	7 (24.1)	19 (27.5)	
>50%	9 (22)	0 (0)	9 (13)	
Procedural success	35 (85.4)	29 (100)	64 (91.4)	0.04
Complete revascularization	25 (61)	26 (89.7)	51 (72.9)	0.01
TIMI flow				
1	4 (9.8)	0 (0)	4 (5.7)	0.04
2	4 (9.8)	0 (0)	4 (5.7)	
3	33 (80.5)	29 (100)	62 (88.6)	
Peri-procedural complications				
Hematoma	2 (4.9)	1 (3.5)	3 (4.3)	0.6
Stroke/TIA	0 (0)	0 (0)	0 (0)	
MI	1 (2.4)	0 (0)	1 (1.4)	0.6
Atrial fibrillation	1 (2.4)	1 (3.5)	2 (2.9)	0.3
Need for external electric shock	2 (4.9)	1 (3.5)	3 (4.3)	0.6
Bradi-arrhytmia	7 (17.1)	2 (6.9)	9 (12.9)	0.3
Death during PCI	3 (7.3)	0 (0)	3 (4.3)	0.3
CIN *	4 (13.3)	2 (9.5)	6 (11.8)	0.5

CIN—contrast induced nephropathy; LM—left main; PCI—percutaneous coronary intervention; SB—side branch; TIA—transient ischemic attack; TIMI—thrombolysis in myocardial infarction. * In 51 patients with pre- and post-PCI creatinine measurements.

**Table 5 jpm-13-00670-t005:** Predictors of early death (30 day) adjusted for cardiogenic shock in patients with unprotected left main coronary artery disease and acute coronary syndrome (Group A—one stent strategy; Group B—two stent strategy).

	HR	CI	*p* Value
Group B vs. Group A	0.04	0.002;0.67	0.03
Post-PCI LAD diameter	0.02	0.0007;0.47	0.02
Post-PCI LCX diameter	0.19	0.05;0.74	0.02
Procedural success	0.03	0.002;0.47	0.001
TIMI flow < 3	0.05	0.005;0.46	0.009

LCX—left circumflex artery; LAD—left anterior descending artery; TIMI—thrombolysis in myocardial infarction; HR—hazard ratio.

**Table 6 jpm-13-00670-t006:** Predictors of late mortality and target lesion revascularization according to the stent strategy in patients with unprotected left main coronary artery disease and acute coronary syndrome (Group A—one stent strategy; Group B—two stent strategy).

	Univariable Analysis			Multivariable Analysis		
4-Year All Cause Mortality	HR	CI	*p* Value	HR	CI	*p* Value
Group B vs. Group A	0.39	0.15;1.01	0.05	0.26	0.09;0.74	0.01
Age	1.03/year	0.99;1.07	0.1			
Male gender	2.87	0.85;9.69	0.09			
SYNTAX	1.08/unit	1.03;1.12	0.001			
SYNTAX-2 PCI	1.07/unit	1.04;1.11	<0.001			
SYNTAX-2 CABG	1.07/unit	1.02;1.11	0.001			
EUROSCORE II	1.13/unit	1.08;1.19	<0.001			
Cardiogenic shock	8.52	3.58;20.2	<0.001	5.62	1.85;17.02	0.002
Atrial fibrillation at presentation	2.89	1.06;7.89	0.04			
LVEF (%)	0.93/%	0.9;0.96	<0.001			
LVEF < 30%	7.89	3.42;18.17	<0.001	3.7	1.35;10.13	0.01
Severe mitral regurgitation	3.86	1.41;10.55	0.008			
Hb before PCI	0.71	0.55;0.9	0.007			
Hb < 12 g/dL	4.07	1.76;9.41	0.001	2.67	1.11;6.42	0.03
Creatinine clearance before PCI	0.97	0.95;0.98	0.001			
CIN *	8.15	2.62;25.27	<0.001			
Associated two/three vessel disease	4.81	1.7;13.57	0.003			
Diffuse LM plaques	2.74	1.12;6.69	0.03	2.85	1.09;7.44	0.03
LM stenosis (%)	1.02/%	1;1.03	0.008			
Post-PCI LCX diameter (mm)	0.39/mm	0.26;0.59	<0.001			
Pre-PCI IABP	4.87	1.76;13.4	0.002			
Post-PCI IABP	2.41	0.89;6.51	0.08			
Predilatation at nominal diameter *	3.27	1.31;8.18	0.01			
POT used	0.43	0.19;1	0.05			
Residual SB stenosis > 50%	3.36	1.23;9.13	0.02			
Procedural success	0.13	0.05;0.37	<0.001			
Complete revascularization	0.23	0.1;0.52	0.001			
	Univariable analysis					
**TLR**	**HR**	**CI**	***p* value**			
Diabetes	12.59	1.46;108;37	0.02			
History of atrial fibrillation	6.57	1.32;32.65	0.02			

* >10% missing values, not used in multivariable analysis. CIN—contrast-induced nephropathy; Hb—hemoglobin; KBPD—kissing balloon post-dilatation; LAD—left anterior descending artery; LM—left main; LVEF—left ventricular ejection fraction; PCI—percutaneous coronary intervention; POT—proximal optimization technique; LCX—left circumflex artery; TLR—target lesion revascularization; HR—hazard ratio.

## Data Availability

Data is unavailable due to privacy or ethical restriction.

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
