# Peer review of "Unprotected Left Main Bifurcation Stenting in Acute Coronary Syndromes: Two-Stent Technique versus One-Stent Technique"

_jpm, 2023, doi:10.3390/jpm13040670_

Round 1
Reviewer 1 Report
This is an interesting study that aims to compare the use of one-stent versus two-stent techniques in patients with unprotected distal left main coronary artery disease (UDLMCAD) presenting as acute coronary syndrome (ACS). The study provides valuable insights into the management of UDLMCAD in the context of ACS and highlights the need for further research in this area.
Author Response
Dear reviewer,
Thank you very much for your sincere appreciation and for your review of our work.
Reviewer 2 Report
Dear Authors,
Your substantial work is to be congratulated; this is a very important topic with very scarce prior evidence and your results and conclusion is well taken.
They are reassuring the evolving concept of "simple is not always better" regarding specifically distal left main disease in this case. However, we have to reckon the small number of patients when we consider your results, thus I would suggest to put somwhere in the Discussion sentence saying that "No harm was detected when 2-stent strategy was implemented when needed.
You depict correctly that adverse outcome was due to suboptimal TIMI flow at post-proc. of the index intervention. But please elaborate on the cases of stent thrombosis in this regard.
My other request: please elaborate on patient "selection": these patients were refused by heart surgeons? (possible due to prior dual antiplatelet therapy?)
Minor comments:
In the Methods: the word "imaging" could be skipped or replaced by angiogram. "confounding": can you rephrase?
2.2 TIMI is Thrombolysis in Myocardial...
Periproc. MI definition should be specified (how increase in CKMB).
Tables are bit "overwhelming", if you can, skip unnecessary parameters. Length in mm
Table 2 needs Legend, add description of LM take-off angle.
3.3 "same moment" instead: same session.
Table 3. Please add info on Fluoro time, Radiation exposure and Contrast used.
Table 5. If you can, replace OR with HR as in Table 6. It would be easier to understand.
Table 6. Can you add "residual Syntax score"?
In the Discussion: "KDPD".
Overall, your work is impressive.
Sincerely
Author Response
Dear reviewer,
Thank you very much for your sincere appreciation and for your detailed review of our work. Your suggestions are very useful. I will detail my revisions and modifications I've made to the manuscript.
I introduced in the discussion a sentence saying that "No harm was detected when 2-stent strategy was implemented when needed”.
All the patients were refused by heart surgeons due to emergency presentation and prior antiplatelet therapy.
There were two cases of acute instent thrombosis and one case of acute side branch vessel thrombosis with fatal outcome. In cases of acute instent thrombosis, no precipitating mechanical factors have been identified for it such as underexpansion or nonapposition of the stent. The most likely cause of these acute instent thrombosis was inadequate antiplatelet therapy. In the case of acute thrombosis at the level of the circumflex artery after left main PCI, we noticed that KBPD was not performed, which could have been the cause of the thrombosis.
The definition of peri procedural myocardial infarction was detailed.
OR was replaced with HR.
LM take-off angle resemble the angle between the left main and the sinus of Valsalva.
Unfortunately, we don't have data about the residual syntax score.
Thank you for your comprehensive review.
Reviewer 3 Report
Dear authors.
Thanks for submission of your work. Basically, I respect your hard work. However, PCI for Lt. main stem bifurcation should be described the technique using stent. Double kissing, Single kissing, Crushing type must be announced for representing result. Please revise the contents adding the detailed technique for 2 stents.
Sincerely
Author Response
Dear reviewer,
Thank you very much for your sincere appreciation and for your review of our work.
We described the two stent techniques that were performed.
In Group B the two-stent techniques used were: provisional T stenting (3.5%), T And small Protrusion (TAP) (35%), minicrush (31%), double kissing crush (3.5%), culotte (27%).